# Exploration of the Interrelationship within Biomass Pyrolysis Liquid Composition Based on Multivariate Analysis

**DOI:** 10.3390/molecules27175656

**Published:** 2022-09-02

**Authors:** Genmao Guo, Qing Huang, Fangming Jin, Linyi Lin, Qingqing Wang, Qionglin Fu, Yin Liu, Muhammad Sajjad, Junfeng Wang, Zhenni Liao, Miao Cai

**Affiliations:** 1Center for Eco-Environmental Restoration Engineering of Hainan Province, Key Laboratory of Agro-Forestry Environmental Processes and Ecological Regulation of Hainan Province, State Key Laboratory of Marine Resource Utilization in South China Sea, College of Ecology and Environment, Hainan University, Haikou 570228, China; 2School of Environmental Science and Engineering, Shanghai Jiao Tong University, Shanghai 200240, China; 3Chenzhou Institute of Forestry, Chenzhou 423000, China; 4Pujin Environmental Engineering (Hainan) Co., Ltd., Haikou 570125, China

**Keywords:** pyrolysis liquid, acetic acids, phenolics, pyrolysis temperature, multivariate statistical analysis

## Abstract

The diverse utilization of pyrolysis liquid is closely related to its chemical compositions. Several factors affect PA compositions during the preparation. In this study, multivariate statistical analysis was conducted to assess PA compositions data obtained from published paper and experimental data. Results showed the chemical constituents were not significantly different in different feedstock materials. Acids and phenolics contents were 31.96% (CI: 25.30–38.62) and 26.50% (CI: 21.43–31.57), respectively, accounting for 58.46% (CI: 46.72–70.19) of the total relative contents. When pyrolysis temperatures range increased to above 350 °C, acids and ketones contents decreased by more than 5.2-fold and 1.53-fold, respectively, whereas phenolics content increased by more than 2.1-fold, and acetic acid content was the highest, reaching 34.16% (CI: 25.55–42.78). Correlation analysis demonstrated a significantly negative correlation between acids and phenolics (r^2^ = −0.43, *p* < 0.001) and significantly positive correlation between ketones and alcohols (r^2^ = 0.26, *p* < 0.05). The pyrolysis temperatures had a negative linear relationship with acids (slope = −0.07, r^2^ = 0.16, *p* < 0.001) and aldehydes (slope = −0.02, r^2^ = 0.09, *p* < 0.05) and positive linear relationship with phenolics (slope = 0.04, r^2^ = 0.07, *p* < 0.05). This study provides a theoretical reference of PA application.

## 1. Introduction

Thermal pyrolysis is one of the most promising ways for the conversion of abundant biomass residues into the biochar, pyrolysis liquid and various volatile gases [1,2].

Pyrolysis liquid (PA), also named pyroligneous acid, is a crude condensate generated from the distillation of smoke produced during biomass pyrolysis, and is highly oxygenated organic smoke liquid, comprising organic acids, phenolics, aldehydes, ketones and alcohols [3,4]. PA can be used as antimicrobial and antioxidant agents to improve plant growth and enhance soil nutrient health conditions because of its advantageous physical and chemical properties [5,6]. The application of PA is attributable to its chemical compositions directly [7,8,9].

The chemical compositions of PA depend on multiple factors influencing pyrolysis (e.g., feedstocks and pyrolysis conditions). When pyrolysis temperatures increased from 300 °C to 600 °C, acids and aldehydes content decreased, whereas phenolics and ketones obtained from giant reeds increased [10]. Wei et al. [11] reported that the PA components were prepared by pyrolyzing walnut shells within three temperatures ranges. The results from their study indicated that the acids content at 151–310 °C was 4-fold higher than that observed at 311–550 °C, whereas the phenolics content was the highest at 151–310 °C, 2-fold higher than that observed at 90–150 °C. The decomposition of wheat straw at 350 °C and 450 °C was studied using the Py-GC/MS technique in a helium atmosphere to determine the gaseous compounds, and the results indicated the formation of compounds classified as phenolic compounds [12].

Furthermore, the chemical compositions of different raw materials PA was different at the identical pyrolysis temperatures. The previous studies have demonstrated that the contents and types of phenolics, acids, aldehydes, lipids and ketones in five kinds of PA obtained from agriculture and forestry wastes were different when carbonization was performed at temperatures of 400–600 °C [13]. Phenolics and acids were the main components in the *Spina* date seed and peanut shell PA when the carbonization temperature was performed at temperatures of 170–400 °C [14]. Multivariate data analysis methods were widely conducted to summarize the process dataset and reduce high-dimension systems [15]. The PCA method can separate the variables to obtain main components for explaining detailed data analysis and it can reveal the inner positive or negative related connections among variables [16]. Correlation analysis refers to the analysis of two or more variables with correlation relationships [17]. However, comprehensive scientific data regarding PA components obtained by multivariate statistics analysis are scarce.

Thus, the aim of this study was to explore the differences in PA components in different feedstocks materials and under different pyrolysis temperatures. Multivariate statistical analysis was conducted using PA compositions data from 42 published papers from 1996 to 2022 along with our experimental study data. This study could contribute to improved guidance on PA preparation, benefiting the comprehensive use of biomass waste via pyrolysis.

## 2. Results and Discussion

### 2.1. Effect of the Feedstock on PA Components

To explore the PA composition difference in the different feedstocks, PCA was not obviously clustered (PC1 variance = 26.72%, PC2 variance = 20.79%) (Figure 1).

The types with the most components, including acids, phenolics, ketones, aldehydes, lipids and others were identified from a total of 20 feedstocks. The relative content of acids and phenolics (the mean and 95% CI) accounted for 58.46% (CI: 46.72–70.19) of the total PA components, which were 31.96% (CI: 25.30–38.62) and 26.50% (CI: 21.43–31.57), respectively, and the relative contents of aldehydes, ketones, alcohols and esters were 6.67% (CI: 4.29–9.05), 7.69% (CI: 6.62–8.77), 5.86% (CI: 4.65–7.06), 3.74% (CI: 0.47–7.00), respectively (Figure 2). Acids production might be ascribed to the decomposition of hemicellulose, cellulose and lignin during the pyrolysis process [18]. Phenolics are produced from lignin decomposition, and acids are generated from hemicellulose and cellulose degradation during pyrolysis [19], ketones are derived from the polysaccharides depolymerization and the monosaccharides isomerization in hemicellulose, and alcohols are attributed to the side chain of aliphatic alcohol hydroxyl breakage in lignin. Lignin, hemicellulose and cellulose decomposition could form aldehydes during the pyrolysis process [20,21]. Wu et al. [22] found that the higher hemicellulose content in bamboo contributed to the highest acids content in bamboo PA, while the lowest acids content in Chinese fir PA was due to its lower hemicellulose content.

### 2.2. Effect of Pyrolysis Temperatures on PA Components

The PA composition difference in two temperatures ranges (below and above 350 °C) was performed by PLS-DA analysis. The PA compositions (PLS1 variance = 27.7%, PLS2 variance = 12.9%) exhibited a significant clustering (Figure 3A). The PLS1 axis could mostly separate two temperatures ranges (below and above 350 °C) of the PA sample. Moreover, the results of the ANOSIM analysis showed that there was significant difference of PA components at two different temperatures ranges (*p* < 0.05). The relative content of acids (43.30%; CI: 34.71–51.89) at below 350 °C was 5.2-fold higher than that at above 350 °C (8.31%; CI: 4.51–12.12); The relative content of phenolics increased by more than 2.1-fold when temperatures ranges increased from below 350 °C to above 350 °C, which were 17.20% (CI: 13.06–21.34) and 36.14% (CI: 26.22–46.06), respectively, while ketones content decreased by more than 1.53-fold from below 350 °C (9.75%; CI: 7.82–11.68) to above 350 °C (6.37%; CI: 3.78–8.97) (Figure 3C). Furthermore, the results of RF models also demonstrated that acids and phenolics were an important predictor in PA constitutes, which was viewed as dominant composition (Figure 3B). In our experimental study, the relative contents of acids in the temperatures ranges 240–270 °C was 2.89-fold higher than that temperatures ranges 400–420 °C, the relative content of phenolic in the temperatures ranges 400–420 °C was 2.29-fold higher than that of temperatures ranges 240–270 °C in the *Eucalyptus* PA compositions. Therefore, this experimental study result was in agreement with the results concluded by data from scientific papers that acids and phenolic contents decreased and increased, respectively, when the pyrolysis temperatures ranges increased from below 350 °C to above 350 °C (Figure 3A). Hemicellulose started to decompose at the lower pyrolysis temperature range (200–320 °C), followed by cellulose degradation at 240–350 °C and lignin at 350–600 °C [23,24]. Lignin decomposition is main source for generating phenolics compounds, whereas the formation of acids, aldehydes and ketones mainly depends on cellulose and hemicellulose decomposition. Therefore, 350 °C was adopted as a demarcation point of the pyrolysis temperature for this study.

### 2.3. PA Compounds Generated during Pyrolysis

A total of 152 chemical compounds were identified in this study of the eucalyptus pyrolysis liquid (Figure 4). In the case of 32 acid compounds, the highest relative content of acetic acid (34.16%; CI: 25.55–42.78) increased by an average of 9.75-fold compared with propionic acid (3.50%; CI: 2.52–4.50). The production of acetic acid is ascribed to the breakdown of the acetyl groups attached to xylan units, resulting from the hemicelluloses dehydration reaction [25]. Acetic acid was formed by the elimination of the carbonyl and O-methyl groups from 4-O-methylglucuronic, the propionic acid formation is attributed to the elimination of the acetaldehyde from the O-acetylxylan unit during the hemicellulose pyrolysis process [26]. The formation pathway of acetic acid was predominant, as acetyl groups accounted for a larger proportion of the feedstock material [27].

The identified phenolics compounds (relative content > 5%) among a total of 35 phenolics compounds were 2,6-methoxyphenol (6.50%; CI: 4.63–8.35), 2-methoxyphenol (7.41%; CI: 4.67–10.15), 2-methlyphenol (5.04%; CI: 3.56–6.52) and phenol (9.34%; CI: 4.57–14.12). The formation of 2,6-methoxyphenol and 2-methoxyphenol are attributed to the vigorous decomposition of the lignin-containing methoxy group [28,29]. The emergence of radical-induced rearrangement and the homolysis of aromatics O–CH_3_ bonds in guaiacyl and syringyl aromatic compounds formed 2-methoxyphenol and 2,6-methoxyphenol [30]. During the further pyrolysis, 2-methoxyphenol and 2,6-methoxyphenol could be susceptible to converted to catechol and phenol, and phenol could be produced from free radical-induced rearrangement reaction as well as from 2-methoxyphenol demethylation [31]. Liu et al. [32] proposed that catechol was transformed by the further guaiacol degradations due to rearrangement and hemolysis, with the methoxy group converting into O-hydroxyphenoxy radical.

A total of 60 ketones compounds were obtained, which was the highest in the PA compounds number in this study. The compounds (relative content > 2%) were 1-hydroxy-2-acetone (3.77%; CI: 2.66–4.87) and 3-methyl-1,2-cyclopentenone (2.94%; CI: 2.28–3.60). The production of ketones is ascribed to the decomposition of hemicellulose and cellulose. First, cellulose tended to depolymerize into active cellulose, in which intermediate products (e.g., levoglucosan) could be produced by the cleavage of β-1,4 glycosidic bonds and intramolecular rearrangement, then the levoglucosan further decomposes to open the furan ring and the cleavage of the C–C bond, followed by1-hydroxy-2-acetone formation [33]. Mansur et al. [34] found that 1-hydroxy-2-acetone could be susceptible to convert into acetone through ketonization. When the temperature increased from 250 °C and 300 °C, hemicellulose initiated polysaccharide chains depolymerization to form oligosaccharides, followed by the cleavage of the xylan chain in the glycosidic linkage to generate 3-methyl-1,2-cyclopentenone.

In this study of the eucalyptus pyrolysis liquid, the furfural (4.30%; CI: 2.76–5.82) and 5-methyl furfural (1.26%; CI: 0.65–1.87) were abundant among the 12 aldehyde compounds. The production of furfural occurs due to the xylose structural unit of the hemicellulose structure, in which hemicellulose undergoes a ring-opening reaction by breaking the bond [35,36].

A total of 23 alcohol compounds were identified in the eucalyptus pyrolysis liquid with the relative content of furfuryl alcohol (2.68%; CI: 1.46–3.89) being the highest. During the pyrolysis process, some C–C bonds in the pyranose ring of glycosides in cellulose were broken and furfuryl alcohol thus could be generated through dehydration reaction [37,38].

### 2.4. Correlation Relationship of PA Compositions

Phenolics, acids, aldehydes, ketones, alcohols, lipids and others were assessed by the correlation analysis. The results demonstrated a significantly negative correlation between acids and phenolics (r^2^ = −0.43, *p* < 0.001) and a significantly positive correlation between acids and lipids (r^2^ = 0.28, *p* < 0.05). In addition, ketones significantly positively correlated with alcohols (r^2^ = 0.26, *p* < 0.05) (Figure 5A).

Acetic acid negatively correlated with 2,6-methoxyphenol and positively correlated with 1-hydroxy-2-acetone. Catechol was positive correlation with 2,6-methoxyphenol. Furfural was positively correlated with 5-methy furfural and 2-methy propanoic acid (Figure 5B).

### 2.5. Linear Relationship between PA Compositions and Pyrolysis Temperatures

The regression analysis showed that pyrolysis temperatures had a significantly negative linear relationship with acids (slope = −0.07, r^2^ = 0.162, *p* = 0.0001) and aldehydes (slope = −0.02, r^2^ = 0.09, *p* = 0.01) and a significantly positive linear relationship with phenolics (slope = 0.04, r^2^ = 0.07, *p* = 0.01) (Figure 6A). Meanwhile, pyrolysis temperatures also had a significantly negative linear relationship with the PA compounds of acetic acids (slope = −0.05, r^2^ = 0.12, *p* = 0.005) and furfurals (slope = −0.01, r^2^ = 0.06, *p* = 0.03) and a significantly negative linear relationship with phenols (slope = 0.03, r^2^ = 0.09, *p* = 0.01) (Figure 6B). These results indicated that pyrolysis liquids components were affected by pyrolysis temperatures during the pyrolysis.

To our best knowledge, this is the first research combining the data from scientific papers from a period of 26 years and using experimental study data to evaluate the effect of temperature and feedstocks to PA components. This study found that temperature significantly impacts PA types and relative content (Figure 3 and Figure 5). In order to better verify this conclusions, we determined eucalyptus PA compositions at two different temperatures ranges (240–270 °C and 400–420 °C) in the experiments. The results showed that there was a significant difference in the PA component at two different temperature ranges when using statistical analysis (Figure 2 and Figure 3). Nevertheless, there were a number of sources of uncertainty and parameters that were not taken into account in this study, mostly due to a lack of data; specifically, oven design, metal ions, heating rate (fast and slow), catalyst, pressure, residence time, different chromatographic conditions (stationary phase, column dimensions, mass spectra identification version, split ratio, etc.) and feedstocks moisture contents. However, these are not insignificant factors to consider when assessing the influencing factors on PA composition. PA was produced in different pyrolysis conditions during data collection in scientific papers. Therefore, these multivariate variable need to be normalized to two variables (temperature and feedstocks) for analysis. These factors may cause partial overlapping to occur in the PA samples (Figure 3A). In this study, PLS-DA analysis was aimed at identifying PA component differences in two groups (below 350 °C and above 350 °C). To further quantitative analysis, random forest (RF) models showed that acids and phenolics were dominant in the composition (Figure 3B).

Lu et al. [39] found that acid content decreased after pretreatments with inorganic acids and increased after pretreatments with organic acids and alkaline compounds, indicating that chemical pretreatment could influence PA components during the biomass pyrolysis. In addition, PA components were influenced by the presence of alkaline metal ions during the pyrolysis. The presence of the metal magnesium impurities likely inhibited the production of ketones. Moreover, GC–MS and mass spectra identification of the chemical compounds was able to influence the peak area of the identified PA components. PA components were different using GC–MS analysis with the capillary column of DB-17MS, AB-FFAP and HP-5MS and different temperature programs [40,41,42,43,44]. Moreover, the boxplots showed that there were significant differences between phenolics, ketones and acids content in two temperatures ranges groups (*p* < 0.05). Previous research aimed to establish the equivalent relationship by replacing high temperature with short residence time in order to explore the influence of residence time on organic components in pyrolysis [45], as pyrolysis liquid components during the pyrolysis reaction are affected by residence time [46]. Due to multiple factors influencing the components during the PA preparation [47]. Interestingly, it is worth pointing out that acetic acid negatively correlated with 2,6-methoxyphenol and positively correlated with 1-hydroxy-2-acetone, while catechol positively correlated with 2,6-methoxyphenol and furfural positively correlated with 5-methy furfural (Figure 5). Hydroxy-2-acetone and acetic acid started to form due to hemicellulose, and cellulose started to decompose at the lower pyrolysis temperature range, followed by lignin degradation at 350–600 °C, while catechol and 2,6-methoxyphenol content increased. Lignin decomposition is the main source for generating phenolic compounds, whereas the formation of acids, aldehydes and ketones mainly depends on cellulose and hemicellulose decomposition. The relative content of the acids decreases and phenolics increases as the temperature increases, and ketone content increases first but decreases gradually with the increasing pyrolysis temperature.

## 3. Materials and Methods

### 3.1. Data Collection

A scientific paper search of peer-reviewed articles published within a range of 26 years (from 1996 to 2022) was conducted using the following seven databases: Web of Science, Scopus, Science Direct, Wiley, SpringerLink, ProQuest and PubMed. The search key words used were wood vinegar, pyroligneous acid, smoke liquid, and pyrolysis liquid. Meanwhile, scientific papers were also searched to prevent omissions, and the research standard included research papers and excluded review articles and book chapters. The scientific papers search strategy was as follows: first, a total of 120,080 documents were screened from seven databases using the four keywords. The initial screening was conducted by reading the title and abstract and resulted in 138 studies. Then, 30 duplicates were removed and further detail screening was performed through the full text. At the end of the process, the initial document list was narrowed to 42 studies.

In our experimental data, the pyrolysis liquid was produced from eucalyptus with a traditional black charcoal kiln and collected by using running water through a shuttle, to condense the smoke, PA were collected from a temperatures range at two different temperatures ranges (240–270 °C and 400–420 °C). The GC–MS analysis of the PA sample was conducted by Shimadzu GCMS-QP2010 Plus (Shimadzu, Japan) at an ionization voltage of 70 eV and an electron multiplier and transfer line temperature of 220 °C on an HP5-MS capillary column (100 m × 0.25 mm i.d, 0.25 µm film thickness). The temperature program was as follows: 40 °C for 2 min, then increased at a rate of 1 °C/min to 70 °C and held for 10 min, was further increased at a rate of 5 °C/min to 260 °C, and was then held constant at 260 °C for 8 min. Other detailed GC–MS parameters were as follows: injection temperature, 200 °C; ion source temperature, 200 °C, carrier gas, Helium at 1.0 mL/min; injection volume 1.0 µL; and mass range m/z 40–500. The identification of chemical organic compounds was based on the comparison of the experimental data with the NIST library database. A flow chart of pyrolysis liquid data from the scientific databases and experimental data, respectively, is depicted in Figure 7.

### 3.2. Data Processing

The PA components data extraction criteria covered included feedstock materials, pyrolysis temperatures, constituent compounds, and relative content (%) according to the final 42 selected studies. The mean value was calculated when the pyrolysis temperature range was present in the data collection. Based on the abovementioned criteria, a list of 162 PA chemical compounds, 20 kinds of feedstock materials and 88 pyrolysis temperature data were generated in total for multivariate statistical analysis. A total of 162 chemical compounds were divided into the seven categories: phenolics, acids, aldehydes, alcohols, lipids, ketones and others. Data for the 88 pyrolysis temperatures data (35–627 °C) were categorized into two groups, with temperature ranges of 35–349 °C and 350–627 °C named as below 350 °C and above 350 °C, respectively, which included 34 sets below 350 °C data and 54 sets above 350 °C data (Appendix A).

### 3.3. Multivariate Statistical Analysis

#### 3.3.1. Principal Component Analysis

Principal component analysis (PCA) is one of the most popular methods in statistical analysis and provides a window into any typical latent structure in a large dataset [48]. The central idea of PCA is to identify a small number of standard or principal components that adequately summarized a large part of the variation of the data and the dimensionality of the problem [49]. PCA, in this study, was used to identify PA component differences at below 350 °C and above 350 °C and of different raw materials. PCA was performed using the software R (3.6.1) with the ggplot2 and vegan packages.

#### 3.3.2. Random Forest Model

Random forest (RF) is a machine learning-based method enabling classification and regression analysis [50], considered to be more accurate classifiers [51]. The importance of the predictor variable was determined by assessing the mean decrease accuracy (MDA) and the mean decrease Gini (MDGini) [52]. In this study, RF classification was used to identify the dominant PA composition between the temperatures ranges above and below 350 °C. The PA composition with significant MDA and MDGini (*p* < 0.05) was assessed by the rfpermute package and were defined as the dominant component. In contrast, insignificant MDA and MDGini (*p* > 0.05) were defined as minor components.

### 3.4. Statistical Analysis

A nonparametric Wilcoxon test was used to determine the PA composition difference between the temperature categories of below 350 °C and above 350 °C. Correlation analysis was conducted using R (3.6.1) with the corrplot package, which is based on the Pearson coefficient. ANOSIM similarity analysis aimed to determine the significance of PA component differences in the temperature categories of below 350 °C and above 350 °C and was conducted using R (3.6.1) with the vegan package. The linear relationship between the pyrolysis temperatures and the PA components were determined by R (3.6.1) with the ‘lm’ function. The relative contents of the PA composition were used to calculate means and 95% confidence intervals (CI) by SPSS 25.0. The heat map was created using R with a heatmap package. PLS-DA (partial least squares discrimination) analysis was conducted using R (3.6.1) with the mixomics package.

## 4. Conclusions

The multivariate statistical analysis of existing published data and our experimental study data showed that the chemical constituents were not significantly different in PA prepared from different feedstock materials. There is a significant linear relationship between the temperature and PA components. There was negative correlation between acids and phenolics substance. Acid and phenolic contents were decreased and increased, respectively, when the pyrolysis temperature range increased from below 350 °C to above 350 °C. Acetic acids were the predominant compounds in the PA chemical compounds. This research provides useful information for PA application in the fields of medicine, food safety and preservation.

## Figures and Tables

**Figure 1 molecules-27-05656-f001:**
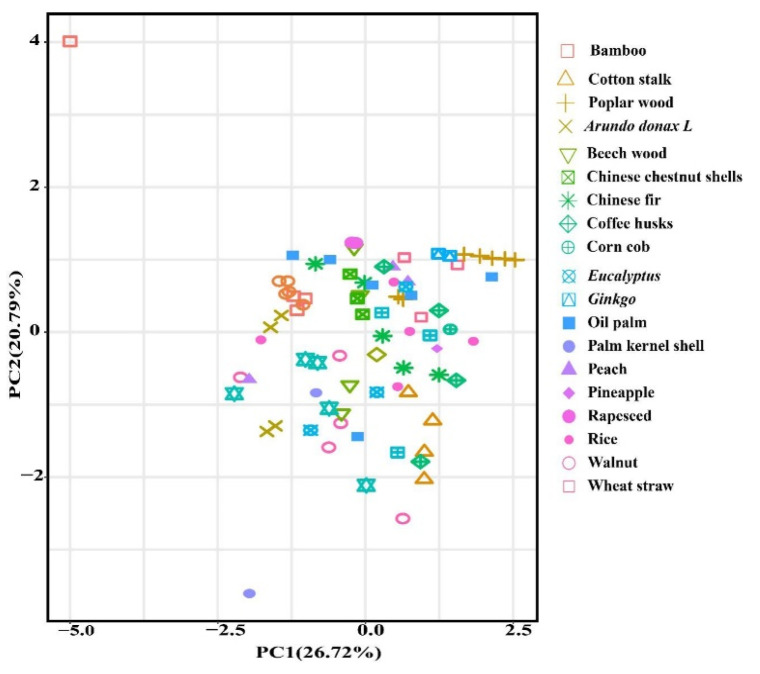
Principal components analysis plot demonstrated pyrolysis liquid from different feedstock.

**Figure 2 molecules-27-05656-f002:**
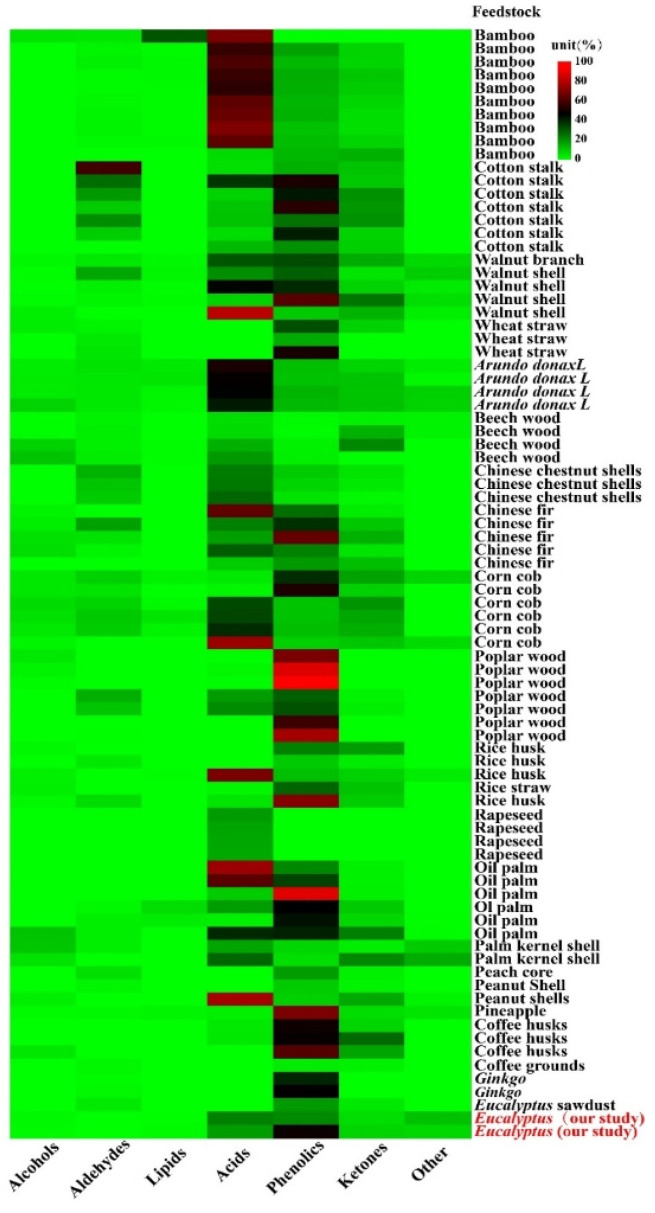
Different feedstock pyrolysis liquid compositions content. The color bar indicated the contents distribution of each pyrolysis liquid compositions (%).

**Figure 3 molecules-27-05656-f003:**
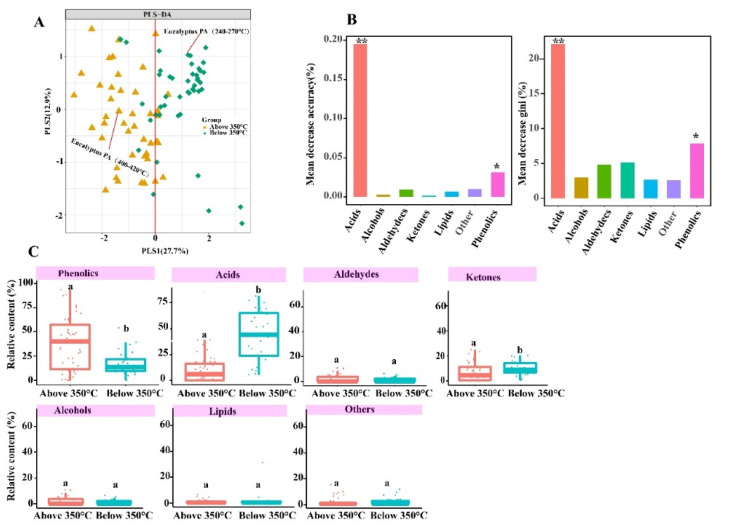
Effect of pyrolysis temperatures on PA components. (**A**) PLS-DA analysis of pyrolysis liquid compositions from different pyrolysis temperatures. (**B**) relative importance of pyrolysis liquid compositions in different pyrolysis temperatures. * indicated *p* < 0.05, ** indicate *p* < 0.01. The out-of-bag error rate of RF model was 12.5%. (**C**) The distribution of phenolics, acids, ketones, alcohols, lipids, aldehydes and others relative content at different pyrolysis temperatures. *p* < 0.05.

**Figure 4 molecules-27-05656-f004:**
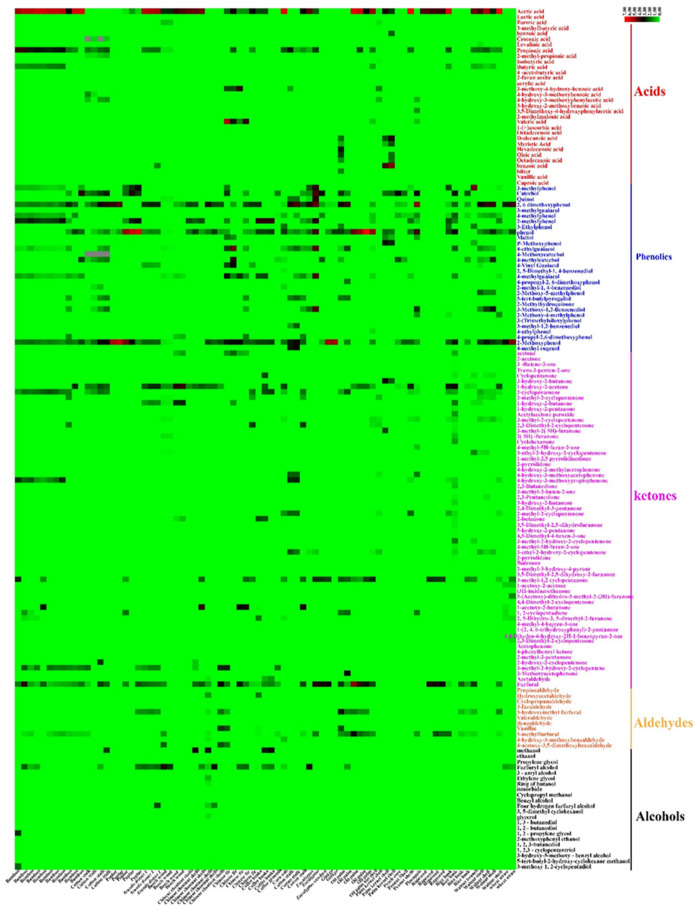
Heat map analyses showing 162 PA compounds content. The color bar stands for the relative content distribution of each wood vinegar composition (%). The red of stands for high relative content, the green stand for low relative content, which the color code grades from green (low content) to black (medium content).

**Figure 5 molecules-27-05656-f005:**
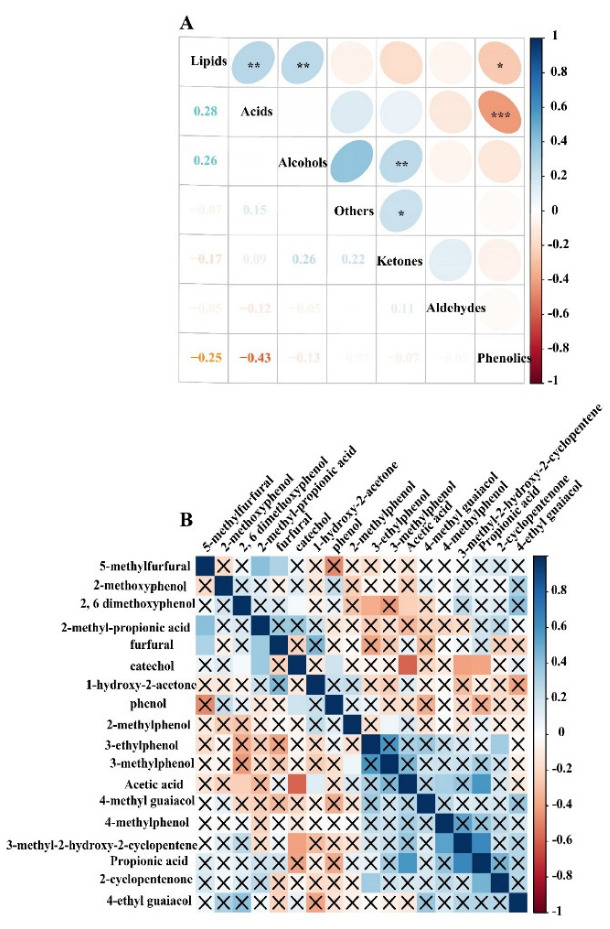
Correlation relationship of PA compositions. (**A**) Correlation analysis for PA compositions phenolics, acids, aldehydes, ketones, alcohols, lipids and others; the color blue of stands for positive correlation, the red stand for negative correlation, the Pearson correlation coefficients are shown in the lower left panel, * indicates *p* < 0.05, ** indicates *p* < 0.01, *** indicates *p* < 0.001. (**B**) Correlation analysis for the relative content of PA compounds for top 20 (relative content ≥ 1%), correlation *p* > 0.05 is indicated by a cross.

**Figure 6 molecules-27-05656-f006:**
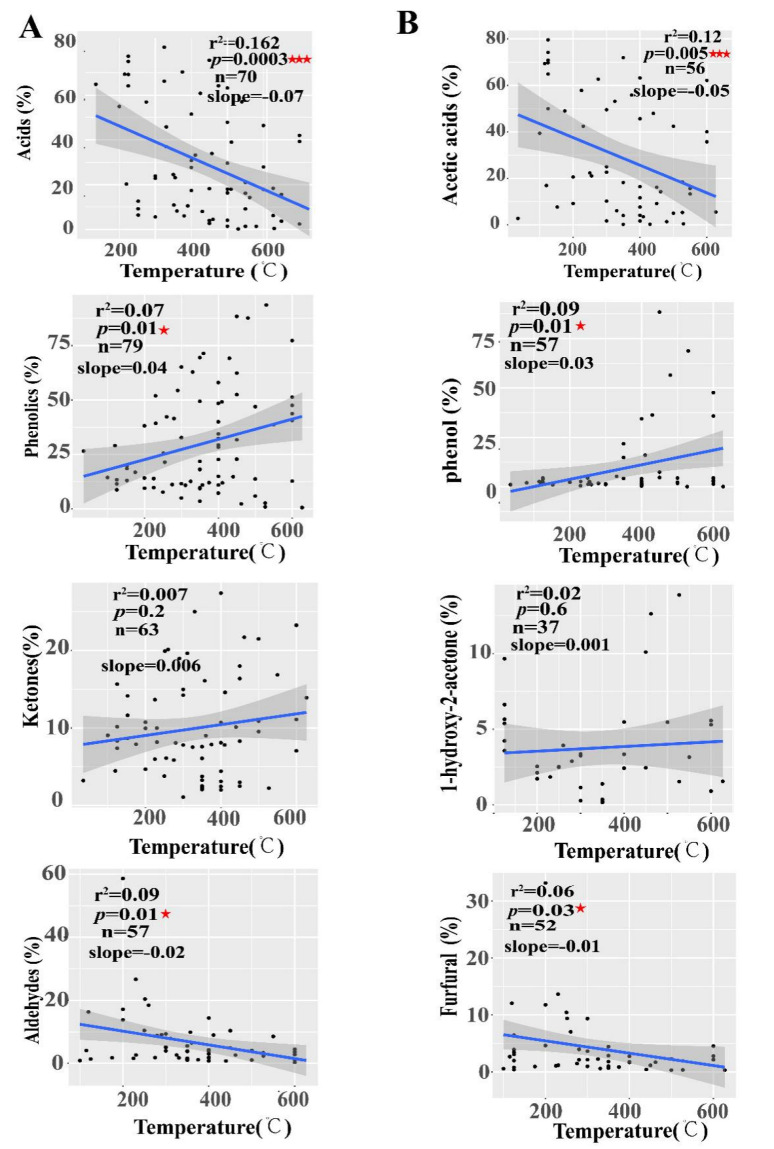
Correlation relationship of PA compositions with pyrolysis temperatures. (**A**) The linear relationship between pyrolysis temperatures and pyrolysis liquid composition types; Main pyrolysis liquids compounds (**B**), ^★^ indicate *p* < 0.05, ^★★★^ indicate *p* < 0.001. The number of data points (n) used for the analysis is given.

**Figure 7 molecules-27-05656-f007:**
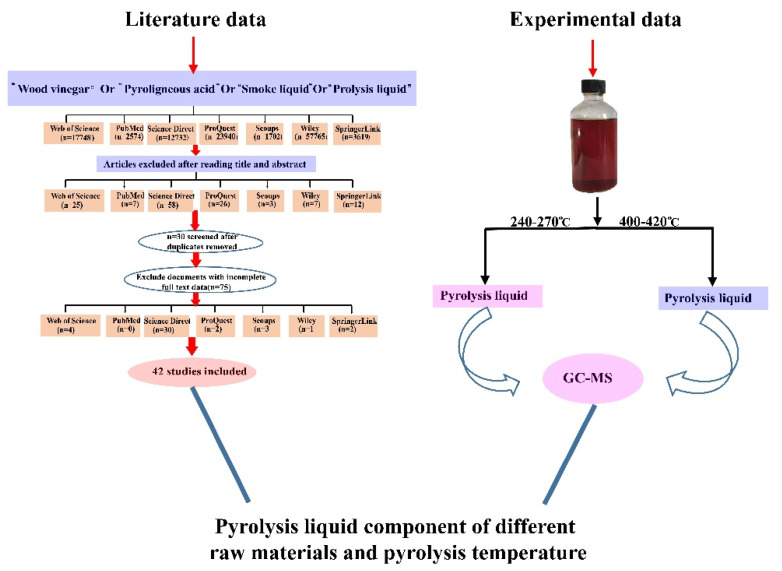
Flow chart of pyrolysis liquid data from the scientific databases and experimental data.

## Data Availability

Not applicable.

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
