# Peer review of "Exploration of the Interrelationship within Biomass Pyrolysis Liquid Composition Based on Multivariate Analysis"

_molecules, 2022, doi:10.3390/molecules27175656_

Round 1
Reviewer 1 Report
The figures are not visible.
As for the characteristics of biomass, the literature review was limited, the authors should take into account similar studies that have been carried out in the discussion. The authors described the subject of biomass in a limited way. I recommend that you refer to biomass in more chemical aspects, I recommend reading the article: https://doi.org/10.3390/pr9020364 .For example in the above article, the PY-GC-MS of straw was shown.
It would be also good to describe the economic impact of the cultivation used and compare how much the commonly used cultivation (an example estimate by the used fertilizers) would cost to that proposed by the authors' cultivation. This is important because the cost of implementing the cultivation is the basis for its application.
Author Response
Dear Editors and Reviewers:
Thank you for your quick response to our manuscript “Exploration of the Inner-relationship among Biomass Pyrolysis Liquid Composition based on Multivariate Analysis” (molecules-1856436). Your comments enlarge our ken, I am sure that are well valuable and very helpful to our researches in future. We have studied the comments carefully and have made correlation which we hope meet with approval. In the following pages are our point-by-point responses to each of the comments of the 1st reviewers.
Response to Reviewer 1 Comments:
Comment 1: As for the characteristics of biomass, the literature review was limited, the authors should take into account similar studies that have been carried out in the discussion. The authors described the subject of biomass in a limited way. I recommend that you refer to biomass in more chemical aspects, I recommend reading the article: https://doi.org/10.3390/pr9020364 . For example in the above article, the PY-GC-MS of straw was shown.
It would be also good to describe the economic impact of the cultivation used and compare how much the commonly used cultivation (an example estimate by the used fertilizers) would cost to that proposed by the authors' cultivation. This is important because the cost of implementing the cultivation is the basis for its application.
Response : Dear reviewer, thank you for your valuable and constructive comment. We have read the article: https://doi.org/10.3390/pr9020364 and Cited in the paper, which described the economic impact of the cultivation used and compare how much the commonly used cultivation (an example estimate by the used fertilizers) would cost to that proposed by the authors' cultivation. The added section as followed:
The decomposition of wheat straw at 350 and 450â—¦C was studied using the Py-GC/MS technique in ahelium atmosphere to determine the gaseous compounds, the results indicated the formation of compounds classified as phenolic compounds [52].
Comment 2: The figures are not visible.
Response : Thanks for your valuable advice. We have added revise high quality of the figures in the paper
Reviewer 2 Report
In this paper, composition of pyrolysis oil was investigated by using multivariate data analysis. Data was collected from literature as well as experimental studies of the authors. The methodology is quite fine and the study gives important findings regarding the interaction of the individual compounds in pyrolysis and pyrolysis temperature. There are still some parts to be revised;
-In introduction section; more emphasis should be made on the importance of multivariate data analysis. literature review should be expanded to highlight the merit of the study.
- The quality of the figures is too low, please revise.
-Sec. 2.1 Author stated that "Phenolics are produced from lignin decomposition, and acids are generated from hemicellulose and cellulose degradation during pyrolysis [15], ketones are derived from the polysaccharides depolymerization and the monosaccharides isomerization in hemicellulose, and alcohols are attributed to the side chain of aliphatic acohol hydroxyl breakage in lignin. Lignin, hemicellulose, and cellulose decomposition could form aldehydes during the pyrolysis process " Is it possible to investigate the relation between content of the individual component and pyrolysis oil?
- The effects of temperature and feedstock type on pyrolysis oil composition was investigated. I believe heating rate should be also taken into account since it considerably affects oil composition.
Author Response
The 2nd Response to review molecules-1856436
Dear Editors and Reviewers:
Thank you for your quick response to our manuscript “Exploration of the Inner-relationship among Biomass Pyrolysis Liquid Composition based on Multivariate Analysis” (molecules-1856436). Your comments enlarge our ken, I am sure that are well valuable and very helpful to our researches in future. We have studied the comments carefully and have made correlation which we hope meet with approval. In the following pages are our point-by-point responses to each of the comments of the 2nd reviewers.
Response to Reviewer 2 Comments:
Comment 1: In introduction section; more emphasis should be made on the importance of multivariate data analysis. literature review should be expanded to highlight the merit of the study.
Response : Thank you for costing your precious time and effort to review our papers. We added more the importance of multivariate data analysis to highlight the merit of the study. The revision section as followed:
As multivariate data analysis methods were widely conducted to summarize the process dataset and reduced high-dimension systems [49]. PCA method can separate the variables to get main components for explaining detailed data analysis and it could reveal the inner positive or negative related connections among variables [50]. Correlation analysis refers to the analysis of two or more variable with correlation relationship [51].
Comment 2: The quality of the figures is too low, please revise
Response : Thanks for your valuable advice. We have added revise high quality of the figures in the paper
Comment 3: Sec. 2.1 Author stated that "Phenolics are produced from lignin decomposition, and acids are generated from hemicellulose and cellulose degradation during pyrolysis [15], ketones are derived from the polysaccharides depolymerization and the monosaccharides isomerization in hemicellulose, and alcohols are attributed to the side chain of aliphatic acohol hydroxyl breakage in lignin. Lignin, hemicellulose, and cellulose decomposition could form aldehydes during the pyrolysis process " Is it possible to investigate the relation between content of the individual component and pyrolysis oil?
Response : Dear reviewer, thank you for your valuable and constructive comment. There is the relation between content of the individual component and pyrolysis oil. Due to limited data collection, we only explored the relationship between temperature and raw materials and components.
Comment 4: The effects of temperature and feedstock type on pyrolysis oil composition was investigated. I believe heating rate should be also taken into account since it considerably affects oil composition
Response : Dear reviewer, thank you for your valuable and constructive comment. heating rate should be also taken into account since it considerably affects oil composition. Due to limited data collection, we only explored the relationship between temperature and raw materials and components. In the next study, we will explore the relationship between heating rate and oil composition.
Round 2
Reviewer 1 Report
The article was well-prepared in accordance with the comments. I recommend this document to publish.